# Internal Validation of a Real-Time qPCR Kit following the UNE/EN ISO/IEC 17025:2005 for Detection of the Re-Emerging *Monkeypox virus*

**DOI:** 10.3390/diagnostics13091560

**Published:** 2023-04-26

**Authors:** Antonio Martínez-Murcia, Aaron Navarro, Adrian Garcia-Sirera, Laura Pérez, Gema Bru

**Affiliations:** 1Department of Microbiology, University Miguel Hernández, 03312 Orihuela, Spain; 2Genetic PCR Solutions™, 03300 Orihuela, Spain

**Keywords:** *Monkeypox virus*, qPCR, detection, validation, UNE/EN ISO/IEC 17025:2005

## Abstract

Human mpox is caused by the *Monkeypox virus*, a microorganism closely related to the Variola virus, both belonging to the *Orthopoxvirus* genus. Mpox had been considered a rare disease until a global outbreak occurred in 2022. People infected with the virus present similar symptoms to patients suffering smallpox and other rash illnesses, hindering diagnosis. The WHO indicated that no commercial PCR or serology kits are currently widely available. In the present study, the MPXV MONODOSE dtec-qPCR kit was validated following guidelines of the UNE/EN ISO/IEC 17025:2005. The parameters evaluated for the acceptance of the assay were in silico and in vitro specificity, quantitative phase analysis, reliability, and sensitivity. The assay passed validation criteria and yielded an efficiency of 95.8%, high repeatability, reproducibility, and a Limit of Detection and Quantification of at least 10 copies. Results from the validation of the MPXV dtec-qPCR kit were satisfactory. The use of the MONODOSE format (dehydrated single PCR-tubes, ready to use) provided considerable advantages allowing the detection of the *Monkeypox virus* to be accurately achieved. This detection kit may be considered a reliable, fast, simple, and universally available option.

## 1. Introduction

Mpox (MPX) is a zoonotic disease caused by the *Monkeypox virus* (MPXV), which belongs to the Poxviridae family and *Orthopoxvirus* (OPXV) genus. The OPXV genus includes viruses such as camelpox (CMPV), cowpox (CPXV), vaccinia (VACV), and variola (VARV), among others. Since smallpox eradication in 1980, MPXV has been the main member of this genus affecting the human population, as declared by the World Health Organization (WHO) [1,2]. This organism was isolated for the first time in the Democratic Republic of the Congo in a patient suspected of suffering from smallpox [1]. So far, two differentiated genetic clades of MPXV have been described: the West African clade, now called clade II (IIa and IIb,) and the Congo Basin clade, currently clade I [3]. Viruses belonging to both groups can infect a wide range of mammals. However, the viruses included in clade II are the source of a milder disease associated with fewer deaths, while clade I presents a higher mortality rate and severe symptoms [1,4,5,6]. Until recently, it was believed that the West African clade presented no human-to-human transmission. Nevertheless, during the 2022 global outbreak, it has been made clear that this clade has the capability to spread between human individuals [7,8,9]. Clinical characteristics of MPXV infection include a short febrile prodromal period, general headache and fatigue, lymphadenopathy, and facial rash that quickly spreads to the rest of the body [1].

MPX has been historically considered an infrequent illness; however, it has become globally relevant due to the increasing number of reports from around the world [4,10]. The proliferation of the virus among the human population has been attributed to respiratory droplets, contact with mucocutaneous lesions of an infected subject, and consumption of contaminated meat [2,11]. In a previously reported MPXV human outbreak in the United States of America in 2003, the causing agent was infected prairie dogs; nevertheless, the origin of the recent outbreak is still unknown [6,11]. Although the virus has been traditionally related to the African continent, it seems that a cross-continent transmission of the virus has been occurring for an unsuspected long time [3]. The 2022 outbreak shows significant differences from previously reported cases; especially important is the predominance of human-to-human infection [6]. The WHO declared MPXV as a Public Health Emergency of International Concern in July 2022. This highlights the current relevance of developing more advantageous detection technologies in order to contain outbreaks [1,12,13]. In the literature, primer and probe sets for PCR assays designed to detect MPXV are available. Nonetheless, it seems they may be improved, either because they target the whole genus or because they lack complete specificity for the selected target [14,15,16,17]. Therefore, the WHO indicated that no commercial PCR or serology kits are currently widely available, although some kits are under development [18,19]. Additionally, the number of sequences in public databases is constantly growing, and the actualization of the MPXV detection methods is required to avoid future issues concerning specificity.

In the present study, the recently developed and worldwide accessible MPXV MONODOSE dtec-qPCR kit (GPS™, Alicante, Spain), commercially available since the end of May 2022, was subjected to analytical validation following the guidelines of the UNE/EN ISO/IEC 17025:2005 with strict acceptance criteria. Validation terms included in silico and in vitro specificity, quantitative phase analysis, reliability, and sensitivity. An external reference laboratory has evaluated the method using clinical samples, and a comparison with several other PCR methods has been performed.

## 2. Materials and Methods

### 2.1. Real-Time PCR Tests

Standard Template, a synthetic double-stranded, linear DNA template reproducing a fragment of the B6R gene and with a known number of copies (10^6^ copies per 5 µL), has been used for the validation of the qPCR. For the quantification of the Standard Template copy number, nanograms of DNA were calculated through spectroscopy, and the copy number was estimated with the exact size of the fragment. The obtained value was confirmed by intercalibration with a qPCR assay. A standard calibration curve was generated by preparing ten-fold dilution series ranging from 10^6^ to 10 copies of the Standard Template provided in the kit. As described by the manufacturer, 5 μL of each dilution of the Standard Template were added to single-dose qPCR tubes provided in the MPXV MONODOSE dtec-qPCR kit (GPS™, Orihuela, Spain) for Positive control. For the Negative control, 5 μL of nuclease-free water was added. The reaction mixtures were subjected to qPCR in a QuantStudio3 (Applied Biosystems, Waltham, MA, USA) device programmed with a standard regime which includes, first, an activation step performed at 95 °C for 2 min, and then 40 cycles of amplification composed by the following steps: denaturation at 95 °C for 5 s and annealing/extension finally fixed at 63 °C for 20 s. All tests include an exogenous Internal control provided in the kit. The main target was read with the FAM channel, while the Internal control was read with the HEX channel.

### 2.2. Sequences Alignment and Phylogenetic Analysis

Sequences from B6R gene (944 bp) for all the species of the OPXV genus (*Monkeypox virus*, *Vaccinia virus*, *Ectromelia virus*, *Cowpox*, *Akhmeta virus*, *Variola virus*, *Taterapox virus,* and *Camelpox virus*) were downloaded from the National Center for Biotechnology Information (NCBI) database [20]. Alignment and phylogenetic analysis were performed using the Mega 11.0.10 software [21] with the neighbour-joining method [22] and bootstrap values for 1000 replicates.

### 2.3. Validation of the Assay following Standard Guidelines

Validation of the method used for the MPXV MONODOSE dtec-qPCR kit was performed as indicated by the guidelines of the UNE/EN ISO/IEC 17025:2005 and by the French Standard NF T90-471:2010. The parameters used for the validation were: in vitro specificity (inclusivity and exclusivity), the analysis of the quantitative phase employing a standard curve calibration of ten-fold serial dilutions, which covered from 10 to 10^6^ DNA copies of the standard DNA template distributed along the other reagents in the kit; reliability (repeatability and reproducibility) and sensitivity (limit of detection, LOD, and limit of quantification, LOQ) with rigorous acceptance criteria. All parameters were evaluated for a minimum of 10 assays [23] and 25 assays in the case of LOD and LOQ.

#### 2.3.1. Specificity of the Assay

The in silico specificity of the primers and probes was assessed during the designing step according to an already described protocol [24]. The appropriate software (BLAST 2.14.0+) was used for the analysis of sequences, available on the website of the NCBI (Bethesda, MD, USA) [25]. The genetic marker targeted by this qPCR was the B6R, the same used in the PCR recommended by CDC-Atlanta [14,16]. Nevertheless, the primers and probe were newly designed and located at positions that improved specificity compared to MPXV sequences available at the NCBI and GISAID repositories until September 2022 [20,26]. Comparison in silico was also achieved with sequences corresponding to all the other OPXV species. Additionally, the exclusivity of the assay was assessed in vitro with qPCR assays using synthetic double-stranded DNA fragments corresponding to the closest phylogenetically related CPXV, VACV, and VARV virus.

#### 2.3.2. Study of the Quantitative PCR Phase

The calibration curve was subjected to a linearity analysis conducted using a ten-fold decimal dilution series (with a range between 10–10^6^ copies) of the Standard Template included in the kit. The whole calibration curve was replicated 10 times. The validation of the standard curve was studied using linear regression in a semilogarithmic graph. In order to achieve this, the following equation was applied: Y = a × X + b, where Y is the cycle threshold (Ct) obtained for each decimal dilution included in the range; X the copy number logarithm; a, slope of the linear regression and b, the cut-off regarding Y axis. Mean values from the fitted regression lines exhibited the results of the slope (a) and the regression coefficient (R^2^). Criteria for acceptance requires that the value of the slope to be found between −4.115 < a < −2.839 and an R^2^ value superior to 0.98. Regarding the validation of the linear model, a Fisher test was performed. Once the test was completed, the model was accepted only if the F derived from the assay (F_assay_) was below the F found in a Fisher distribution table (F_fisher_) for (v1 = k−1; v2 = k × [*n* − 1]), with a 95% Confidence Interval. Efficiency (E) was computed in accordance with the equations where the slope was achieved from the linear regression (a): E = 10 ^−1/slope^; e = % Efficiency = (E − 1) × 100. If 75% < e < 125%, then Efficiency could be considered acceptable.

#### 2.3.3. Reliability of Analysis

Reliability is the capability of a method to deliver results without the presence of random errors. This parameter can be evaluated by making use of the assessment of repeatability and reproducibility. The results calculated for both parameters should present coefficient of variation (CV) values of <10% to be deemed satisfactory. The method’s repeatability was assessed by preparing 10 replicates of the ten-fold standard dilution series (from 10^6^ to 10 copies belonging to the Standard Template). Independent tests for each replicated dilution series were performed. The CV was calculated using the following formula: CV = S/$\bar{x}$ × 100, where S is the standard deviation and $\bar{x}$ the average of Ct values. For reproducibility, the values used were attained from two experimental sets, each composed of five standard calibration curves (*n* = 5) prepared by two different technicians on separate dates. The CV was estimated to assess the reproducibility as follows, where Sab is the standard deviation between technician a and b, and $\bar{x}$ab is the Ct value averaged from data sets obtained from technician a and b: CV = Sab/$\bar{x}$ab.

#### 2.3.4. Limit of Detection (LOD) and Limit of Quantification (LOQ)

In order to accept the LOD of the PCR, a verification of the minimum number of target units gives rise to a positive amplification with a 90% confidence level achieved. The LOD of the qPCR in accordance with the laboratory’s procedure test was roughly calculated by performing 15 tests, each composed of 10 copies of the Standard Template. The assessment of the LOQ involves the calculation of the smallest number of target units able to generate a repeatability result of quantification. Therefore, the evaluation of the LOQ of the method giving rise to a repeatability result was accomplished by analysing the results with a *t*-Student test with a Confidence Interval of 95%. Assays were carried out for 10 copies of the Standard Template, which were replicated 15 times. To calculate the experimental *t* value, the results of quantification were used. The calculation of the experimental *t* value is performed as follows, where $\bar{x}$ is the median of the sample, μ is the reference value of copy number, s is the typical deviation, and *n* is the quantity of samples: *t* = ($\bar{x}$ − μ)/(s/√*n*). The precision of the LOQ was acceptable only if the *t* value obtained from the assays was inferior to the theoretical value obtained from the Student table (*t* value < *t*-Student; freedom degree *n* − 1).

## 3. Results

The MPXV MONODOSE dtec-qPCR kit was designed and produced by May 2022, being one of the first commercially available kits when WHO declared MPXV a Public Health Emergency of International Concern in July 2022. After this process, MPXV-specific primers and probe sequences targeting B6R were periodically compared to OPXV sequences available at GISAID and NCBI up to September 2022. These in silico analyses indicated that the design has a high specificity, matching all MPXV sequences and keeping enough nucleotide differences to discriminate the other OPXV species. Nevertheless, the comparison of MPXV showed a significantly high level of similarity to the pair of CPXV sequences of strain Finland 2000 MAN HQ420893 and strain Austria 1999 HQ407377. Therefore, to ascertain the species relationships within the OPXV genus, a phylogenetic analysis including B6R sequences representing the greatest possible diversity of available data was performed. The obtained phylogenetic tree is shown in Figure 1. This tree confirms that sequences from strains Finland 2000 MAN and Austria 1999 showed a very close relationship with the MPXV cluster, although the most common CPXV sequences found in the databases had a much more distant relationship. Consequently, these sequences were used for in vitro exclusivity testing due to the low number of nucleotides mismatching to the qPCR primers and probe; synthetic templates used were selected to contain: i, the sequence corresponding to CPXV strain Finland, HQ420893.1, the closest to MPXV; and ii, the sequence LR800245.1 representing the clinically relevant VARV species.

Despite having verified the in silico exclusivity of the qPCR for most CPXV strains (89 sequences), the in vitro assay with the CPXV strain Finland template displayed a delayed amplification (Ct = 37; 10^4^ DNA copies) when the annealing/extension step was performed at 60 °C (Figure 2). These 2 strains may have been misclassified or be a result of a polyphyletic origin of the CPXV species (Figure 1). In fact, they share an identical amplicon sequence with 11 VACV strains (out of 134 total sequences available in public databases). This implies results obtained for the CPXV strain Finland and Austria should be the same as the qPCR outcome for 11 VACV strains since they share the exact amplicon sequence.

To increase the astringency of the PCR at the primer-hybridization step, the annealing temperature of the assay was increased gradually. Still, a weak amplification was observed when performed at both 61 °C and 62 °C (Ct = 39 and 41, respectively; 10^4^ DNA copies). Finally, at 63 °C, no signal was detected (no amplification), and the final protocol was modified accordingly in the kit manual, changing the annealing/elongation temperature to 63 °C. The Positive controls at increasing temperatures did not suffer substantial modifications, yielding the same Ct values as before the modification of the temperature (Figure 2).

Validation of a set of qPCR reactions using ten-fold serial dilutions from 10 to 10^6^ DNA copies of Standard Template was performed at 63 °C. Ct values obtained from the Positive controls were used to plot a standard curve to estimate the assay’s efficiency (Figure 3). Negative control was performed by substituting the Standard Template with nuclease-free water, and no amplification was observed. The exogenous Internal control showed amplification between Ct 30.9 and 29.9, values included in the range stipulated by the manufacturer.

Following the guidelines of the UNE/EN ISO/IEC 17025:2005, empirical validation terms were assessed for a minimum of 10 assays (25 in the case of LOD and LOQ), and results were subjected to established acceptance criteria (Table 1). Standard curve for calibration was performed using ten-fold serial dilutions of 10–10^6^ from the provided synthetic Standard DNA copies, and the obtained slope (a) and coefficient (R^2^) values were −3.426 and 1.000, respectively; these results are included within the acceptance ranges. The values were considered acceptable since F_assay_ (1.081) was found below the F_fisher_ (5.318), and efficiency e = 95.8%. The CV values to assess the reliability of the method ranged from 0.69% to 3.90% for repeatability and 0.59% to 3.77% for reproducibility; therefore, the method is repeatable and reproducible. LOD and LOQ (the sensitivity) were evaluated for a set of 25 assays containing 10 copies of the Standard Template. LOD was 100% reproducible. The accuracy of LOQ was acceptable since the *t* value obtained from the assays (*t* value = 0.169) was lower than the theoretical value from the Student table (*t*-Student = 2.064).

## 4. Discussion

MPX was first described in 1970 (Congo) and has since spread to several African regions [27,28]. It became a disease of public health relevance in 2003 when an outbreak in the USA occurred [29]. In 2022, more than 114,000 MPX cases were reported in 111 different countries. Currently, the United States of America has reported the most cases, with a number exceeding 30,000; Spain is the third most affected country worldwide, taking the lead in Europe with 7514 confirmed cases, followed by France with more than 4000 reported cases [30,31]. The population that is predominantly vulnerable to the virus is composed of those born after the cessation of routine smallpox vaccination due to the lack of protection granted by cross-immunity [2,32]. Recently, the WHO has declared MPXV as a Public Health Emergency of International Concern [33].

Some primers and probes had been published to enable the detection of MPXV with a PCR assay, and they were reviewed in the present study. A PCR test for the generic detection of OPXV was designed to be inclusive for all the species belonging to the genus [15]. Concerning MPXV-specific detection, some authors proposed a probe designed for the B6R target, which presented a mutation in the 5′ end of 30% of sequences of the MPXV currently available, hindering the obtention of acceptable results [16]. Subsequent studies performed with GR2 as a target showed complete inclusivity; nevertheless, primers and probe sequences were not completely exclusive for MPXV [14]. These studies were an excellent foundation for MPXV detection, but improvement could be achieved by developing more exclusive primers and probes. Last September, while drafting the present article, CDC announced a laboratory alert indicating that some molecular laboratory-developed tests may lead to false negative results. Said tests had been designed using the CDC published primers and probes that specifically detect MPXV; therefore, this event may be caused by some deletions of the tumour necrosis factor (TNF) receptor gene that is present in some specimens [34]. Given recent events, a fast, affordable, simple, sensitive, and reliable method to detect MPXV specifically was advisable to help monitor the expansion of this virus [35,36].

In the present study, in silico analysis performed when reviewing the specificity of the MPXV qPCR kit validated exhibited complete inclusivity for all the MPXV sequences available in the public databases. Results also manifested enough mismatches to discriminate VARV and other OPXV species; nonetheless, a significantly high level of similarity to CPXV B6R sequences (Finland 2000 MAN HQ420893 and Austria 1999 HQ407377), as well as to part of VACV sequences was revealed. Therefore, a phylogenetic analysis including sequences representing the greatest diversity of each OPXV species found in the databases was performed.

This analysis indicated that all OPXV species, except CPXV, grouped at clearly separated and robust clusters, as high bootstrap values were obtained in most cases (Figure 1). However, the relationships displayed in the resulting phylogenetic tree proved that CPXV species, as currently described, comprises several distantly related phylogenetic clusters, suggesting a polyphyletic origin and/or a possible misclassification. This issue was already highlighted by the International Committee on Taxonomy of Viruses (ICTV) in the report for the family Poxvididae, but no further information was accessible [37]. A deeper evaluation of available sequences showed similar relationships from nine housekeeping gene sequences of the same strains selected from databases when a Multi-locus Phylogenetic Analysis (MLPA) was performed (study in progress). This MLPA approach also agrees with a previous study performed using full genomes [38] that was recently reviewed and confirmed using the most conserved core of the genomes [39]. In all, these findings suggest that the current method for identification of the CPXV may lead to confusion regarding taxonomic classification. Closely related sequences from VACV and CPXV (Figure 1) and clinically relevant VARV sequences were selected to construct synthetic templates for in vitro validation of exclusivity. As shown in Figure 3, weak PCR signals and delayed amplifications obtained in the case of 2 CPXV strains and 11 VACV strains, which share a common template, were solved by increasing astringency at the annealing step of the PCR cycling. Finding a solution was a notable improvement of the kit because the previous results may be interpreted as false positives.

The results obtained from the validation of the MPXV MONODOSE dtec-qPCR kit were optimum according to the criteria for acceptance recommended by the guidelines of the UNE/EN ISO/IEC 17025:2005 and by the French Standard NF T90-471:2010. The results and the criteria for each term of validation are summarized in Table 1. To perform quantification experiments of quality, the evaluation of the quantitative PCR phase inside the range of the standard curve is essential. For the validation process, the assay should be repeated a number of times to be statistically relevant; therefore, during the present study, it was considered appropriate to replicate the analysis at least 10 times. The Fisher test was applied to evaluate the linear model, and it was found acceptable since F_assay_ was found to be significantly below F_fisher_. The performance of the method can be evaluated through Efficiency, which was found to be e = 95.8%; even though values above 75% are usually considered acceptable, it is preferred to obtain a value above 90% to ensure the yield of the PCR amplification is appropriate. The method was found repeatable and reproducible because the CV was lower than 10% in all cases and, consequently, reliable. Also, the sensitivity was assessed during the validation of the method since it indicates the ability of the kit to correctly detect the target. Despite the LOD requiring positive results over 90%, this study provided positive outcomes in a 100% of the cases. Finally, the LOQ result was accepted because it was found within the stipulated range, as the *t* value obtained was lower than the *t*_student_ (0.169 < 2.064). In conclusion, the results of the analysed parameters stipulated in the ISO17025 were all accepted according to the criteria established (Table 1).

The MPXV MONODOSE dtec-PCR kit contains individual ready-to-use tubes which include all the components needed for the specific detection of this pathogen performing a qPCR test. This method presents considerable benefits with respect to other PCR methods previously reported [14,15,16,17,40]. This technology is very manageable and straightforward since all the reagents required are dehydrated together, enabling technicians to add their samples and run the PCR without the need to carry out any intermediate steps. Another advantage of this kit format is that the use of dry ice during shipment is not necessary, as it can be transported at room temperature, considerably reducing the time required for delivery and cost. As no freeze-thawing is required, enzyme stability is not compromised. Also, cross-contamination and fluorophore deterioration by UV light has been reduced in the largest amount possible.

A diagnostic validation using clinical samples was conducted by the reference health laboratory Instituto de Salud Carlos III (Madrid, Spain) [41]. During this external study, several commercially available kits (GPS™, ThermoFisher, Roche, MaterLab, Integrate DNA Technologies, NovaTec, Nzytech) and other published PCR methods for the detection of MPXV [14,15,17] have been compared. The diagnostic performance was assessed by comparing the capability of the different available assays to detect the minimum number of copies in a reaction tube, hereby determining the LOD with clinical samples for each methodology. This comparative study also includes the analysis of the diagnostic parameters (diagnostic sensitivity and diagnostic specificity) for MPXV detection using a selection of 40 clinical samples (lesion exudate, skin injury, vesicular fluid), 21 positive and 19 negatives. The MPXV MONODOSE dtec-qPCR kit developed by GPS™ exhibited a 100% for both diagnostic sensitivity and specificity, and, which is most important, it showed the lowest LOD when compared with the other techniques [41]. As previously mentioned, sensitivity is quite an influential parameter during the validation of a method, and the validation described in this study proved the kit to be highly sensitive; this information, together with the results obtained by the ISCIII, suggests that this kit may be suitable for the analysis of human clinical samples. Further testing will be undertaken with the aim of using it as an In Vitro Diagnostic tool (IVD).

In conclusion, considering all the previously discussed results, it appears that the MPXV MONODOSE dtec-qPCR kit may be an eligible candidate to be deemed a robust method for the detection of MPXV. This worldwide available kit might be considered as reliable, sensitive, and innovative, making the method for MPXV detection simple, fast, and safe to use. In sight of the promising results obtained by the ISCIII in the previously described study performed with 40 clinical samples, further research could be conducted with a higher number of clinical samples in order to validate this kit for human diagnosis in the near future.

## Figures and Tables

**Figure 1 diagnostics-13-01560-f001:**
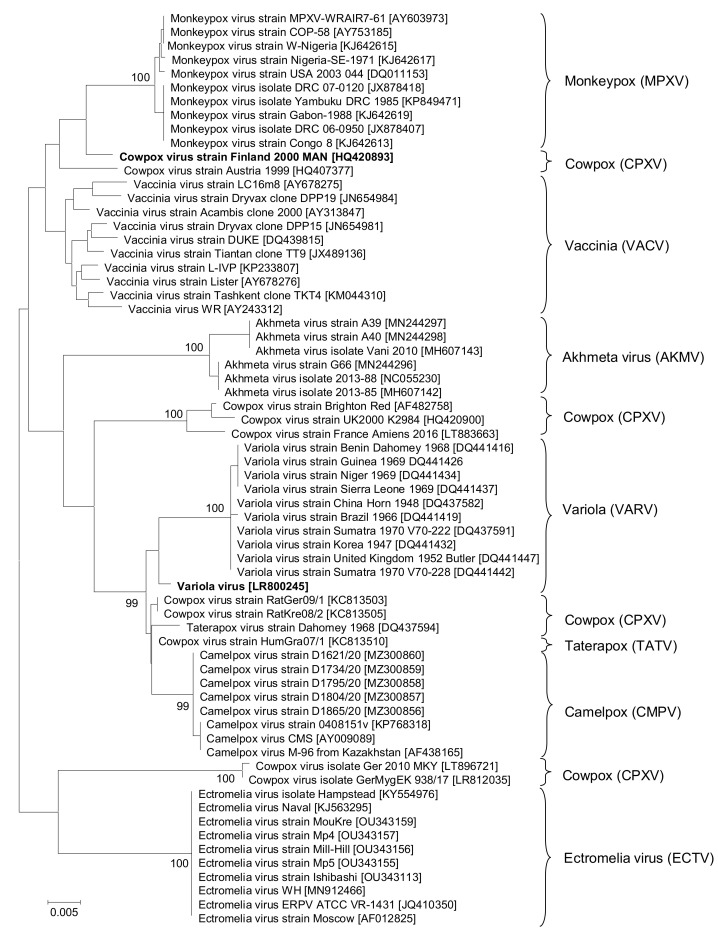
Phylogenetic Neighbour-Joining tree showing relationships of MPXV and the most related species of the OPXV genus. The analysis was derived from the alignment of 944 bp belonging to the B6R gene. Numbers at nodes indicate bootstrap values in percentage for 1000 replicates.

**Figure 2 diagnostics-13-01560-f002:**
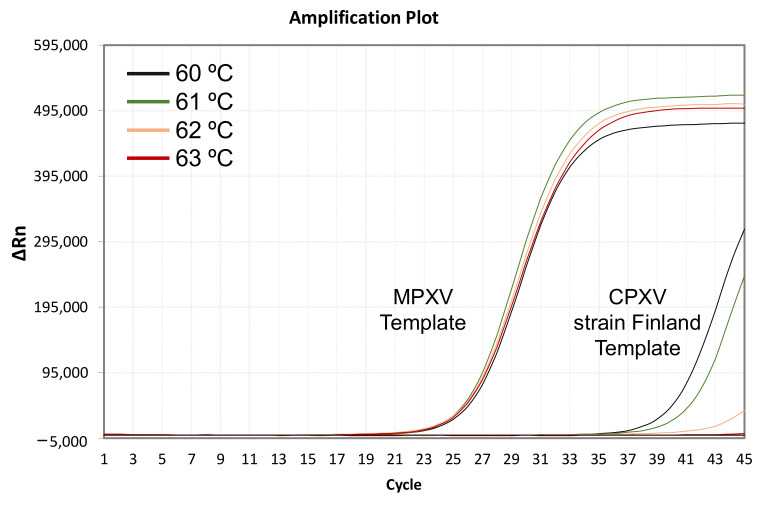
Amplification plots from qPCR assays using synthetic DNA templates containing 10^4^ copies corresponding to Cowpox virus (CPXV) strain Finland HQ420893.1 and 10^4^ copies of *Monkeypox virus* (MPXV) at different annealing temperatures: 60 °C (black), 61 °C (green), 62 °C (orange) and 63 °C (red).

**Figure 3 diagnostics-13-01560-f003:**
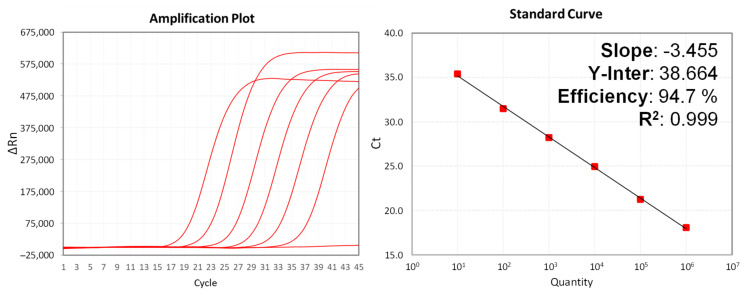
Quality Control of MPXV MONODOSE dtec-qPCR Test using ten-fold serial dilutions of Standard Template from 10 to 10^6^ DNA copies and negative control. Amplification plot (**left**) and calibration curve with statistical parameters (**right**).

**Table 1 diagnostics-13-01560-t001:** Summarized results from the estimation of the qPCR parameters by following standard guidelines of the MPXV MONODOSE dtec-qPCR. Values, criteria for acceptance, and results are included. n, repetitions for each parameter.

Term of Validation	Obtained Values		Acceptance Criteria	Result
Standard curve *n* = 10	Y = −3.426 × X + 38.904 A = −3.426 R^2^ = 1.00		−4.114 < a < −2.839	ACCEPTED
	F_assay_ = 1.081 F_fisher_ = 5.318		F_assay_ < F_fisher_	ACCEPTED
	Efficiency (*e*) = 95.8%		75% < *e* < 125%	VALIDATED
Reliability *n* = 10	Repeatability		CV < 10%	REPEATABLE
	Conc.	CV (%)		
	10^6^	1.10		
	10^5^	0.80		
	10^4^	0.69		
	10^3^	0.78		
	10^2^	1.00		
	10^1^	1.52		
	5	3.90		
	Reproducibility		CV < 10%	REPRODUCIBLE
	Conc.	CV (%)		
	10^6^ copies	0.91		
	10^5^ copies	0.67		
	10^4^ copies	0.59		
	10^3^ copies	0.86		
	10^2^ copies	1.16		
	10^1^ copies	1.64		
	5 copies	3.77		
Detection limit (LOD) *n* = 25	10 copies	Positive = 25/25 (100%)	Positives ≥ 90%	ACCEPTED
Quantification limit (LOQ) *n* = 25	10 copies	*t* value = 0.169	*t* value < *t*_student_	ACCEPTED
		*t*_student_ = 2.064		

## Data Availability

All data have been included in the text.

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
