# Peer review of "Internal Validation of a Real-Time qPCR Kit following the UNE/EN ISO/IEC 17025:2005 for Detection of the Re-Emerging Monkeypox virus"

_diagnostics, 2023, doi:10.3390/diagnostics13091560_

Round 1

Reviewer 1 Report (Previous Reviewer 2)

Dear Authors,

it's a pity for the pic of the virus.

However, in my opinion, the paper can be accepted in the present form.

Author Response

We are very sorry we are unable to include the requested image. 
Thank you for the review and time invested.

Reviewer 2 Report (Previous Reviewer 3)

The authors have improved the manuscript and addressed the queries.

Author Response

Thank you for the time invested in the review and helpful comments

Reviewer 3 Report (New Reviewer)

Since the aim of the work is the validation of the PCR test, it would be very relevant to provide more information a synthetic DNA template (section 2.1 in material and methods). And how the copy number is calculated, this would be very useful for readers interested in reproducing the method.

By what criteria is it mentioned that the specificity of the primer and probe used has been improved? please provide more information

Author Response

Please, find attached the document with our response. 

Reviewer 4 Report (New Reviewer)

The manuscript is suitable to publish in ‘Diagnostics’.

Author Response

We would like to thank you for your review.

Reviewer 5 Report (New Reviewer)

In the manuscript entitled “Internal validation of real-time qPCR kit following the UNE/EN ISO/IEC 17025:2005 for detection of the re-emerging Monkeypox virus” the authors describe, guided by UNE/EN ISO/IEC 17025:2005 recommendations, a thorough internal validation process of a MONODOSE dtec-qPCR kit for the detection of mpox virus. This kit proves to be specific, sensitive and has an acceptable low limit of detection and quantification with good reproducibility, ease of use and promising clinical results therefore fitting to be used as an diagnostic tool. 

However, the following issues are to be addressed before publication

1.       A suggestion that the title be change to “Internal validation of a real-time qPCR for the detection of mpox virus.”

2.       A very untidy and unpolished draft was sent for review. Previous review suggestions were not edited. This needs to be corrected. Parts of the manuscript was not easy to read nor understand. The language usage needs to be improved substantially. a clean copy of the manuscript would be preferable

3.       The WHO has recommended a name change for monkeypox. Please change to new name mpox throughout manuscript eg change line 8 to read “Human mpox is caused by mpox virus….”

4.       Line 8: Change to  ….”an orthopoxvirus”. As it is used in this sentence it should be in lower case letters and not in italics.  However, as referred to it as a genus as in line 25, is correct.

5.       Line 26: different pox viruses should not start with a capital letter.

6.       Line28: smallpox without a capital letter. Amend throughout manuscript

7.       Line 33: Amend to ” ….however, viruses belonging to clade II is……”. A clade cannot be the source of a disease.

8.        Line 55: Change to “….technologies in order to contain outbreaks.”

9. in section 2.1 line 78-81 reference is made to a positive control. However, the validation was done on synthetic standard template which is the control and no additional positive control was used. Kindly clarify.

10. It would be beneficial to highlight the CPXV HQ4208893 sequence on the tree- in bold or colour and also show the VARV representative strain (LR800245.1) on the tree in Figure 1. It is currently not on the tree.

11. The results section would benefit from a better description of exactly what was done and what the outcomes were, in its current format it is difficult to follow. For example,

13. In the results section, the description of the results obtained with the synthetic CPXV and VACV strains lines 210-222, is unclear and needs to be revised. Not sure what is meant by the sentence “ this implies ….in lines 220-222. Were the 11 VACV strains tested?

14.       Line 76-77: Change to …..”A standard calibration curve was generated by preparing a ten-fold dilution……”

15.   Line 82: Change to “was” and not “were” added.

16.   Line 75-88: In this section it would be worth mentioning the gene region detected by this kit

17.   Line 94: All pox virus names should be in small letters. Only when referring to the specie name is it in capital letter and italics.

18.   Line 220: indicate whether these 134 sequences are available in the public domain

19.   Line 220: Clarify in which gene region the 2 questionable CPXV sequences are identical to the VACV viruses compared. If this is the B6R region, then these two CPXV sequences should belong to the VACV cluster on your B6R phylogenetic tree.

20.   Lines 226-230: please clarify

21.   In result section, indicate a PCR cut off value for this PCR kit the authors validated.

22.   Line 322: Change to: “This was a significant improvement since previous results….”

23.   Line 334: Replace “ requested in” with “ as stipulated in “

24.   Line 352: Replace dried-ice with dry ice.

25.   Line 354: Replace with “ As no freeze thawing is required, enzyme stability is not compromised.”

26.   Line 370: Check whether reference 38 is relevant here.

27.   Line 371: A suggestion to change this sentence to read something like this – “ Our validation proved this kit ( give name) to be highly sensitive and together with clinical outcomes obtained by (give Institute name) suggests that this kit may be suitable…….”

28.The R2 value in the table and description of the results is different- 0.999 vs 1. Presumably the one has been rounded off, consistency would be better. In table 1 the percentage positives for the LOD is not given, please include. And there is typographical error “posit”

29.Reference for the statement in lines 307-309 is not given.

30. In lines 357-361, where reference to other publications , the current presentation of the data, reads like this work was part of the current study, please revise to clarify.

31. The authors should state that a major limitation of this work is that it was conducted only on synthetic template and that for full validation the kit needs to be evaluated on isolates/ clinical samples . the specificity should also be confirmed on isolates of closely related viruses- CPXV etc.  other orthopoxviruses.

32. The manuscript requires English language revision- sentences are not grammatically correct- examples listed below. The manuscript will benefit from a review by a native English speaking colleague.

Line 25-29 sentence is not complete/ grammatically correct- requires revision

Line 46-48 sentence beginning “ in a previous .. requires revision

Line 52, the word contamination should be replaced by transmission or infection.

Line 119- doble is misspelt- should be double

Lines 123-125 – please revise the wording it is unclear

Line 175-176 – sentence needs grammatical correction

Author Response

Please, find attached our response to your comments. 

This manuscript is a resubmission of an earlier submission. The following is a list of the peer review reports and author responses from that submission.

Round 1

Reviewer 1 Report

This study just includes analytical validation with clinical validation missing. LOD study just used 15 samples rather 20 samples. Specificity is not tested using real samples. With so many key steps missing, this study is considered too preliminary to be published.

Author Response

Please, find attached the answer to your comments. 

Reviewer 2 Report

Dear Authors, in my opinion, the paper is very well written. As an advice, I would add a figure of the Monkeypox Virus and an appendix of abbreviations.

Author Response

We are grateful for your comments and the time invested in the review.

We are afraid we do not have a picture of MPXV virus produced by us. An in-depth review of the manuscript has been carried out so that all abbreviations are clear throughout the document.

Reviewer 3 Report

In the manuscript entitled “Internal validation of real-time qPCR kit following the UNE/EN ISO/IEC 17025:2005 for detection of the re-emerging Monkeypox virus “by Antonio Martínez-Murcia et al., the team validated the commercial kit MPXV MON-58 ODOSE dtec-qPCR kit (GPS™, Alicante, Spain), by using the guidelines of the UNE/EN ISO/IEC 60 17025:2005. Even though the study is at right time, where there is a need for diagnostic kits against the emerging Monkeypox virus, however, the study lacks the rigor and comprehensive evaluation of the kit. The study lacks novelty.

Major points:

1.     In the Insilco analysis, the MPXV showed a close relationship with Cowpox viruses, however, the qPCR showed negative results. The authors explained this is might be due to the exclusive specificity of MPXV to the kit. However, the explanation is not convincing.

2.     The authors should have validated the kit with MPXV virus DNA from infected human samples. Without the actual results from infected samples, it is hard to claim the sensitivity and specificity.

3.     In the materials and methods section the details of the positive and negative controls and DNA details should have been mentioned.

4.     The quantity of DNA used in the qPCR was not mentioned.

5.     What is the range of Ct value for this kit to be considered as positive or negative for the unknown samples? Please explain.

6.     There is no information on the primers and probes used in this study. Please submit the details of primers and probes.

7.     The results of the kit show that the limit of detection for the DNA samples is reproducible, whether these DNA samples have been checked with other probe-based available kits as a standard in the study.

Author Response

(The authors gave the same response as above.)
